# CausalESC: Breaking Causal Cycles for Emotional Support Conversations with Temporal Causal HMM

## Abstract

Emotional Support Conversation (ESC) is a rapidly advancing task focused on alleviating a seeker's emotional distress. The intricate interplay between cognition, emotion, and behavior presents substantial challenges for existing approaches, which often struggle to capture the dynamic evolution of the seeker's internal state during conversations. To address this, we propose **CausalESC**, a model designed to dynamically represent the seeker's internal states, by assuming that the generative process governing the mutual influence among these factors follows a first-order Markov property, with *i.i.d.* random variables. The model comprises a prior network, that disentangles the seeker's emotions, cognition, and behavior, and a posterior network, which decouples the support strategy factors. The prior network also models the psychological causality of the seeker within each conversation round. To account for the varying effects of support strategies on the seeker's intrinsic states, we incorporate a support intervention module to capture these impacts. Additionally, a holistic damping transfer mechanism is designed to regulate the complex interactions among cognition, emotion, behavior, and strategy, ensuring that changes remain within a reasonable range. Our model effectively breaks causal cycles and achieves causal representation learning. Both automatic and human evaluations demonstrate the effectiveness of our model, emphasizing the advantages of modeling the evolution of the seeker's internal state under support strategies.

## 1 Introduction

In recent years, mental health problems have become increasingly prevalent, yet access to professional psychological counselors remains limited and costly. Consequently, there is an urgent need for a chatbot capable of alleviating psychological issues. Emotional Support Conversation (ESC) (Liu et al., 2021) is designed to reduce individuals' distress by generating appropriate support strategies, as illustrated in Fig. 1. This task holds significant potential in various fields, including mental health support and social assistance.

An increasing number of researchers are investigating ESC tasks. For instance, MISC(Tu et al., 2022) uses COMET to infer the fine-grained mental state of the seeker and then employs a mixture strategy to generate emotional support text. Similarly, GLHG(Peng et al., 2022) also utilizes commonsense knowledge to model local and global hierarchical relationships. Additionally, PAL(Cheng et al., 2023) leverages the seekers' persona to generate more informative and personalized responses. Meanwhile, TransESC (Zhao et al., 2023b) constructs a state transition graph to model semantic, strategy, and emotional transition, thereby generating effective responses.

While previous studies have attempted to model the psychological state of the seeker, they often fall short by representing it as static snapshots, neglecting the continuous and dynamic evolution of the seeker's internal state throughout the conversation. Cognitive Behavioral Therapy (CBT)(Rothbaum et al., 2000) emphasizes the importance of understanding the dynamic relationship between the individual's internal state and their environment during the therapeutic process, as well as the interaction mechanisms among emotions, cognition, and behavior. This concept is illustrated in the left subfigure (Fig.2). Drawing inspiration from CBT, this work models the dynamic evolution between

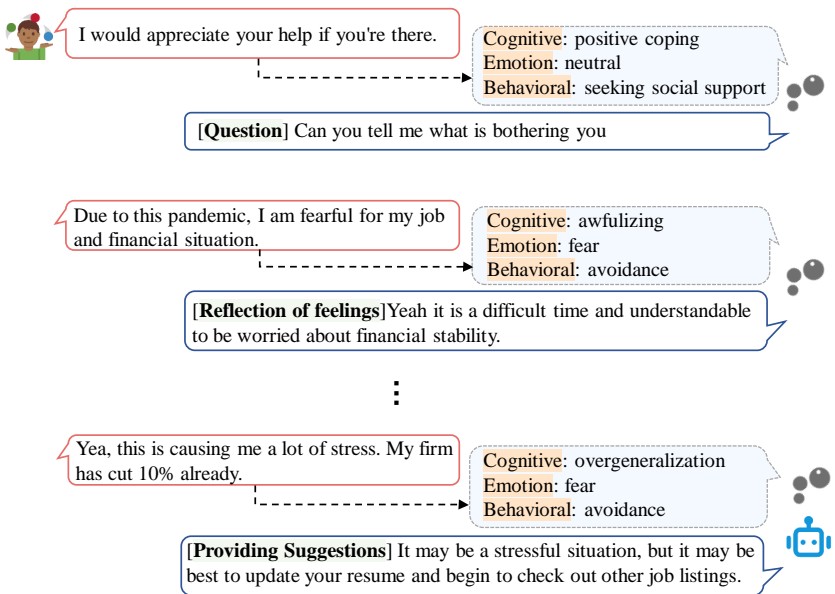

Figure 1: An example of an emotional support conversation is presented, featuring a seeker and a supporter delivering a supportive response.

cognition, emotion, and behavior during emotional conversations. However, traditional causal assumptions typically rely on a Directed Acyclic Graph (DAG), and these cycles present challenges for conventional causal representation learning in capturing the complex relationship in cognition, emotion, and behavior(Forré & Mooij, 2020).

In this paper, we assume that the generative process involving the mutual influence of emotion, behavior, and cognition adheres to the first-order Markov property, with the random variables being *i.i.d.*. Based on this assumption, cognition, emotion, and behavior unfold along the temporal dimension, forming a DAG that satisfies structural requirements. Consequently, we design a causal graph (as shown in the right sub-figure (Fig.2)) to represent the operation of the emotional support dialogue and define a joint distribution (as detailed in Formula 2) to describe the underlying principles governing the generation of observed embeddings.

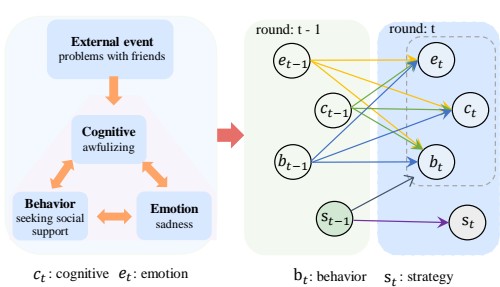

Figure 2: The left figure illustrates the interaction between an individual and their environment, highlighting how this interaction impacts emotions, cognition, and behavior. The right depicts a causal Markov model unfolding over time, resolving the causal loop problem.

Building on this framework, we propose a Temporal **Causal** Hidden Markov Model for the **ESC** task, referred to as **CausalESC**, to model the dynamic evolution of the seeker's internal state under varying supporter strategies and guide the generation of supporters' responses. Specifically, the prior network first utilizes the seeker's utterance as the observation variable, conditioned on the emotion label, to disentangle the seeker's emotional, cognitive, and behavioral factors. Given the stability of psychological mechanisms, the causal relationships between emotions, cognition, and behavior are modeled at each time step. Meanwhile, the posterior network decouples the support strategy factors. Considering that the supporter employs different strategies to support the seeker at each time step, we introduce a strategy intervention approach to dynamically capture the influence of the supporter on the seeker's internal states enhancing the supporters' responsiveness to seekers' needs. Additionally, to model the complex interactions among cognition, emotion,

behavior, and strategy, a holistic damping transfer mechanism is introduced. This mechanism regulates these interactions after each time step, ensuring that changes in the variables remain within a reasonable range. Finally, the causal endogenous variables and strategic factors from the final time step are inject into the decoder to generate the supporter's response. Experimental results from both automatic and human evaluations demonstrate the superiority of our approach. The main contributions of our work are as follows:

- To the best of our knowledge, this paper is the first to learn causal representations within causal loops, resolving circular causality by assuming that the generative process governing the mutual influence of emotions, behaviors, and cognitions follows the first-order Markov property with *i.i.d.*.
- By expanding cognitive, emotional, and behavioral factors into a directed acyclic graph (DAG), each dialogue round influences subsequent generations. Based on this, we propose the CausalESC model to disentangle these causal representations and dynamically capture the evolution of the seeker's internal state, thereby guiding supporters in generating responsive outputs.
- Extensive experiments on benchmark datasets demonstrate that CausalESC is a highly competitive approach for Emotional Support Conversation.

## 2 RELATED WORK

**Emotional Support Conversation** Since being proposed by (Liu et al., 2021), the ESC task has gathered significant attention. MISC(Tu et al., 2022) integrates commonsense knowledge and employs mixed-strategy to guide response generation. GLHG(Peng et al., 2022) utilizes a graph-based reasoner to model the hierarchical relation between global cause and local intention, capturing the multi-source information. SUPPORTER(Zhou et al., 2023) formulates ESC as a process of eliciting positive emotion and designs a mixture-of-expert-based mechanism with a reinforcement learning approach. TransESC(Zhao et al., 2023b) focuses on the fine-grained turn-level transition of ESC, including semantics, strategy, and emotion transition. KEMI(Deng et al., 2023) retrieves mental health knowledge from a pre-trained knowledge graph and evaluates the model from the perspective of mix-initiative. MFF-ESC(Bao et al., 2024) perceives emotional intensity transitions and proposes an information network that integrates text semantics, feedback, and emotional intensity streams. However, all the aforrmentioned methods model the seeker's state only statically. In contrast, we disentangle the seeker's emotion, cognition, and behavior to simulate the dynamic evolution of the seeker's internal state during the conversation.

**Causal Disentangled Representation Learning** In recent years, causal disentangled representation learning has garnered increasing attention from researchers, with its primary objective being the discovery of high-level causal variables from low-level observations (Schölkopf, 2022). Most approaches in this field combine Structural Causal Models (SCM)(Pearl, 2009) with deep learning. For example, CausalVAE (Yang et al., 2021) employs a causal layer to transform exogenous variables into causal endogenous factors that correspond to causally related concepts in data. (Li et al., 2021) present a novel causal hidden Markov model for sequential medical images for future disease prediction. (Zhao et al., 2023c) analyze physical factors in multimodal traffic flow and proposes a causal propose causal conditional hidden Markov model to predict traffic flow. For dialogue data, (Su et al., 2024) propose a temporal causal disentanglement model that effectively decouples the dialog content and realizes the temporal accumulation of emotions, enabling more accurate emotion recognition. The concepts in the above causal disentangled learning are all directed acyclic, but in our task, cognition, emotion, and behavior are interdependent, presenting a challenge for causal representation learning. In our model, we assume the mutual influence of emotion, behavior, and cognition follows a first-order Markov process with *i.i.d.*, effectively resolving the issue of circularity.

## 3 METHODOLOGY

The overall architecture of our proposed approach is illustrated in Fig. 3. Our model is composed of three components: Dialogue Floor Encoder, Temporal Causal Hidden Markov Module and PsychoCausal Hybrid Decoder.

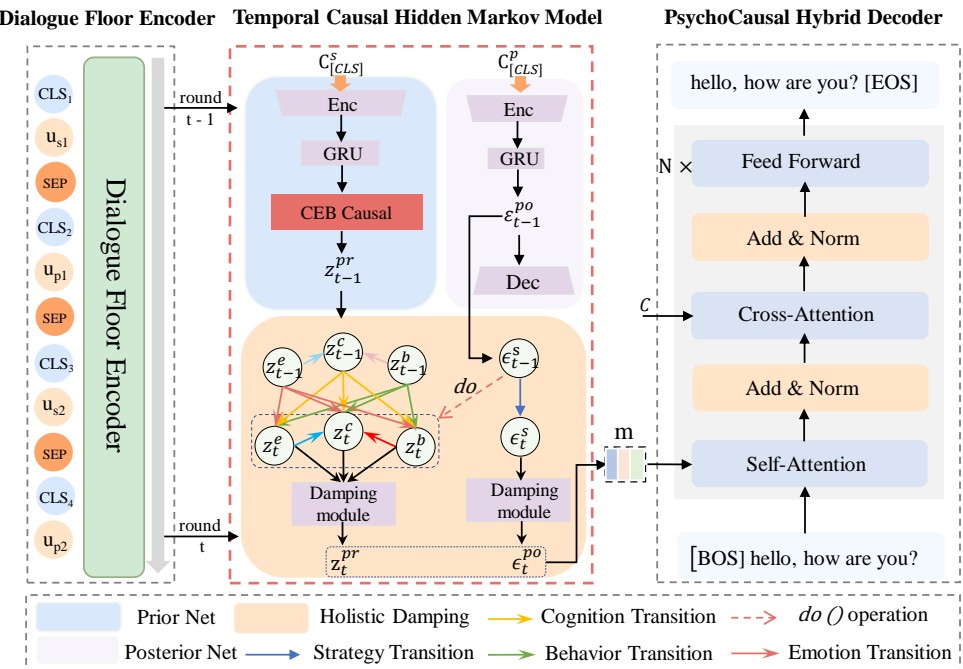

Figure 3: The architecture of CausalESC.

## 3.1 PROBLEM DEFINITION

The ESC task involves a dialogue designed for seeking emotional support and help, where the seeker and supporter speak alternately. Formally, let the dialogue between the seeker $s$ and the supporter $p$ contain $T$ rounds, represented as $U = [u_1^{(s)}, u_1^{(p)}, ..., u_T^{(p)}]$, where $u_i = (w_1^i, w_2^i, ..., w_M^i)$. Here, $u_i^{(s)}$ and $u_i^{(p)}$ denote the utterances of the seeker and the suppoter, respectively. Our goal is to use the conversation history $U$ to generate an appropriate support response $u_T^{(p)}$.

## 3.2 DIALOGUE FLOOR ENCODER

The Dialogue Floor Encoder serves as the semantic context encoder, sharing the same architecture as the encoder used in BlenderBot (Roller et al., 2020), which is pre-trained on large-scale dialogue corpora. Specifically, each utterance is separated by $[SEP]$, and special tokens $[CLS]$ and $[SEP]$ are added at the beginning and end of each sentence in the conversation history, respectively. The encoder $E$ is then employed to encode each word $w$, thereby obtaining its contextual representation.

$$C = E([CLS], u_1, [SEP], [CLS], u_2, ..., [CLS], u_n) \tag{1}$$

where $C$ means conversation context representation. Additionally, we use $C_{[CLS]}$ to represent the embedding representation of each sentence in the conversation.

## 3.3 TEMPORAL CAUSAL HIDDEN MARKOV MODEL

To address the issue of circularity inherent in the causal assumptions, we propose that the generative process governing the mutual influence of emotion, behavior, and cognition satisfies the first-order Markov property with independent and identically distributed (*i.i.d.*) variables. This assumption enables us to model the progression of the conversation as a DAG, as illustrated in Fig. 3. A detailed proof of the acyclic of this model is provided in the Appendix ***Acyclic Proof***.

At each time step $t$, to ensure identifiability(Khemakhem et al., 2020), the seeker's emotional information is combined with the causal endogenous latent variables $z_{t-1}$ from the previous step. This

process facilitates the disentanglement of the independent exogenous variables $\epsilon = [\epsilon_t^c, \epsilon_t^b, \epsilon_t^e]$. Using a SCM, these independent exogenous variables are transformed into causal endogenous variables $z_t = [z_t^c, z_t^e, z_t^b]$. These variables represent approximations of the seeker's internal states, where $z_t^c$, $z_t^e$, and $z_t^b$ correspond to the cognitive state, affective state, and behavioral state of at time step $t$, respectively. Similarly, the supporter's strategy information at time $t$ is integrated with the latent variable $s_{t-1}$ from the previous time step to extract the strategy latent variables $\epsilon_t^s$ at the current time $t$.

### 3.3.1 A Probabilistic Generative Model for CausalESC

According to the causal Markov codition(Pearl, 2009), the joint distribution of the latent variables can be factorized based on the DAG structure derived from the Markov assumption as follows:

$$p_\theta\left(U_{\leq T}^{(p)}, z_{\leq T}, \epsilon_{\leq T}, s_{\leq T}\right) = \prod_{t=1}^{T}\left[p_\theta\left(z_t, \epsilon_t \mid z_{t-1}, s_{t-1}\right) \cdot p_\theta\left(s_t \mid s_{t-1}\right) \cdot p_\theta\left(U_t^{(p)} \mid s_t\right)\right] \quad (2)$$

where the first term represents the prior network, denoting the distribution of the exogenous variable $\epsilon_t$ conditioned on the latent variable $z_{t-1}$ and the strategy $s_{t-1}$ from the previous time step. This distribution can be further factorized into the generation mechanism of exogenous and endogenous variables based on the causal relationships:

$$p_\theta\left(z_t, \epsilon_t \mid z_{t-1}, s_t\right) = \prod_{t=1}^{T}\left[p_\theta\left(\epsilon_t \mid z_{t-1}, s_{t-1}\right) \cdot p_\theta\left(z_t \mid \epsilon_t\right)\right] \quad (3)$$

The second term denotes the transiton probablity of the policy state. The third term refers to the generative model, which characterizes the distribution of the supporter's observed variable $U_{\leq T}^{(p)}$ at the current time step, conditional on the current supporter's strategy $s_t$.

$$p\left(U_{\leq T}^{(p)} \mid s_{\leq T}\right) = \prod_{t=1}^{T} p\left(U_t^{(p)} \mid s_t\right) \quad (4)$$

Since the true posterior is difficult to handle, a tractable distribution $q_\phi$ is constructed to approximate the true posterior $p_\theta$ defined as follows:

$$q_\phi\left(\epsilon_{\leq T}, z_{\leq T} \mid U_{\leq T}^{(s)}, s_{\leq T}\right) = \prod_{t=1}^{T}\left[q_\phi\left(\epsilon_t \mid z_{t-1}, U_t^{(s)}, s_t\right) \cdot q_\phi\left(z_t \mid \epsilon_t\right)\right] \quad (5)$$

### 3.3.2 Prior Network

To establish the prior distribution, a prior network is defined within the model. Traditionally, a standard multivariate Gaussian prior has been commonly employed, but it may limit its ability to handle complex data. To address this limitation, we introduce a prior network into the model to enhance its representation capability. The prior network learns the prior distribution $p_\theta\left(z_t, \epsilon_t \mid z_{t-1}, s_t\right)$, which consists of a GRU module, a support intervention module, and a CEB causal module.

*1) GRU Module:* We use the GRU to model the evolution of the internal state of the seeker. Specifically, we input the observed variable, the seeker's utterance $U$, and the hidden variable from the previous time step into the prior network. A GRU is then employed to propagate the hidden variable across time steps using its single-step dependency mechanism. Finally, the output is passed through two fully connected layers (FCs), with one layer outputting the mean and the other outputting the logarithmic variance.

*2) Support Intervention Module:* During the conversation, the supporter offers guidance to the seeker, influencing the seeker's internal state. We introduce a novel component, termed the Support Intervention Module, to model this infulence. Specifically, the influence of the supporter on the seeker is denoted as $do()$, as illustrated in Fig. 3. In our work, the $do()$ operator is implemented using an attention mechanism, which focuses on both the prior at the current time step and the posterior of the previous step. The process is described as follows:

$$\tilde{\epsilon}_t^{pr} = do(\epsilon_{t-1}^{po}, \mathbf{z}_t^{pr}) = \text{softmax}\left(\epsilon_{t-1}^{po}\mathbf{W}_{att}(\mathbf{z}_t^{pr})^T\right)\mathbf{z}_t^{pr} \quad (6)$$

**3) CEB Causal Module:** According to cognitive psychology, the causal relationship between cognition, emotion, and behavior is a stable psychological mechanism. To this end, we propose a Cognitive-Emotional-Behavioral (CEB) causal module to model thses relationship in each dialogue round. Considering the complex nonlinear relationships between how emotions drive behavior and how cognition influences emotions, we adopt a general nonlinear SCM to represent these intricate interactions. In this paper, the CEB Causal Module is expressed as follows:

$$\mathbf{z}_t^{pr,i} = f_i\left(\left[\left(\mathbf{I} - \text{sigh}\left(\tilde{(}\alpha\mathbf{A})\right)^T\right)^{-1}\epsilon_t^{pr}\right]_{[:,i,]}\right) \tag{7}$$

where $\mathbf{A} \in \mathbf{R}^{3\times3}$ represents the adjacency martrix of DAG, The hyperparameter $\alpha$ is introduced to accelerate convergence. Subsequently, fully connected layers are employed to obtain the mean and log variance.

### 3.3.3 POSTERIOR NETWORK

The purpose of the posterior network is to learn a variational posterior distribution, denoted as $q_\phi\left(\epsilon_{\leq T}, z_{\leq T} \mid U_{\leq T}^{(s)}, U_{\leq T}^{(p)}, s_{\leq T}\right)$, which approximates the true posterior distribution of the latent variables. This network is constructed based on the observed data from the supporter's utterances. Unlike the prior network, the posterior network consists solely of GRU modules to model the evolution of the supporter's strategy. Subsequently, an FC layer is applied to derive the mean and log-variance vectors of the hidden variables.

### 3.3.4 GENERATION NETWORK

In the generation network, the updated hidden variables are used to reconstruct the observed variables. In each dialogue turn, these variables are parameterized by a FC layer to reconstruct the supporter's utterance $U^{(p)}$.

### 3.3.5 HOLISTIC DAMPING TRANSFER MECHANISM

In our model, considering the complex interactions relationships between cognition, emotion, and behavior, we design a holistic damping transfer mechanism that integrates a multi-dimensional interaction transfer mechanism with a damping module.

At each time step $t$, the state of each latent variable $z_t = [z_t^e, z_t^b, z_t^b]$ is influenced by the states of all variables from the previous step and by the perturbation $\epsilon_t^{po}$. The *multi-dimensional interaction transfer mechanism* can be mathematically expressed as:

$$z_{t+1}^e \sim f_e(z_t^e, z_t^c, z_t^b, \epsilon_t^{po}, \eta_{t+1}^e)$$
$$z_{t+1}^c \sim f_c(z_t^e, z_t^c, z_t^b, \epsilon_t^{po}, \eta_{t+1}^c) \tag{8}$$
$$z_{t+1}^b \sim f_b(z_t^e, z_t^c, z_t^b, \epsilon_t^{po}, \eta_{t+1}^b)$$

where $f(\dot{)}$ represents the function that governs the interaction between varibles, $\eta_{t+1}^e, \eta_{t+1}^c$, and $\eta_{t+1}^c$ are *i.i.d.* random variables.

Inspired by emotion regulation theory (Gross, 2002), there is a buffering process in the transmission of internal states such as emotions. Therefore, we introduce a *damping module* to model this regulation mechanism. This ensures that the changes in variables remain within a reasonable range. The damping module is computed as follows:

$$\mathbf{z}_t^{pr} = tanh(\mathbf{z}_t^{pr}) * \sigma(\mathbf{z}_t^{pr})$$
$$\epsilon_t^{po} = tanh(\epsilon_t^{po}) * \sigma(\epsilon_t^{po}) \tag{9}$$

where $\sigma$ represents the sigmoid activation.

### 3.4 PSYCHOCAUSAL MEMORY DECODER

The final latent variable $z_T = [z_T^e, z_T^c, z_T^b]$ and the latent representation $\epsilon_T^{po}$ for the supporter are utilized to guide the text generation. Inspired by (Li et al., 2020), we apply the memory schema to

incorporate this knowledge into the decoder. Specifically, the states of both seekers and supporters are attended to each self-attention layer. The computation is as follows:

$$K = [\mathbf{z}_h, \mathbf{H}^e]\mathbf{W}^K \quad V = [\mathbf{z}_h, \mathbf{H}^e]\mathbf{W}^V \tag{10}$$

where, $\mathbf{z}_h = [[\mathbf{z_t^e}, \mathbf{z_t^c}, \mathbf{z_t^b}], \mathbf{s_t}]$, $H^e$ is encoded context, $\mathbf{W}^K, \mathbf{W}^V \in \mathbf{R}^{d_h \times d_h}$ are the projection matrices.

## 3.5 LEARNING METHOD

Given a dataset $\mathcal{D}$, the Evidence Lower Bound (ELBO) is represented as follows:

$$\mathcal{L}_{\text{ELBO}} = E_{\mathcal{D}}\left[\mathbf{E}_{q_\phi}\left[\log\left(\frac{p_\theta\left(U_{\leq T}^{(p)}, z_{\leq T}, \epsilon_{\leq T}, s_{\leq T}\right)}{q_\phi(z_{\leq T}, \epsilon_{\leq T}, s_{\leq T}, U_{\leq T}^{(p)})}\right)\right]\right] \tag{11}$$

Expand and break it down into individual time steps as follow:

$$\mathcal{L}_{\text{ELBO}} = \mathbf{E}_D\left[\sum_{t=1}^{T}\mathcal{L}_{q_\theta,p_\phi}^t\right] \tag{12}$$

where:

$$\mathcal{L}_{q_\phi,p_\theta}^t = \mathbf{E}_{q_\phi}\left[\log p_\theta\left(U_t^{(p)} \mid s_t\right)\right] - \text{KL}\left[q_\phi\left(\epsilon_t, z_t \mid z_{t-1}, U_t^{(p)}, s_t\right)\|p_\theta\left(\epsilon_t, z_t \mid z_{t-1}, s_t\right)\right] \tag{13}$$

Due to one-to-one correspondence between $\epsilon$ and $z$, we can utilize the Dirac delta function $\delta(\cdot)$ to reformulate both the prior and posterior distributions. the variational posterior is as follows:

$$q_\phi\left(\epsilon_t, z_t \mid z_{t-1}, U_t^{(p)}, s_t\right) = q_\phi\left(\epsilon_t \mid z_{t-1}, U_t^{(p)}, s_t\right) \cdot \delta\left(z_t = \phi\left(\epsilon_t\right)\right)$$
$$= q_\phi\left(z_t \mid z_{t-1}, U_t^{(p)}, s_t\right) \cdot \delta\left(\epsilon_t = \phi^{-1}\left(z_t\right)\right) \tag{14}$$

$$p_\theta\left(\epsilon_t, z_t \mid z_{t-1}, s_t\right) = p_\theta\left(\epsilon_t \mid z_{t-1}, s_t\right)\delta\left(z_t = \phi\left(\epsilon_t\right)\right)$$
$$= p_\theta\left(z_t \mid z_{t-1}, s_t\right)\delta\left(\epsilon_t = \phi^{-1}\left(z_t\right)\right) \tag{15}$$

where $\phi$ is an invertible function. We substitute the prior and posterior distributions from formulas 14 and 15 into formula 13, resulting in the variational lower bound $\mathcal{L}_{q_\phi,p_\theta}^t$ as follows:

$$\mathcal{L}_{q_\phi,p_\theta}^t = \mathbf{E}_{q_\phi}[\log p_\theta\left(U_t^{(p)} \mid s_t\right)]$$
$$- \text{KL}\left[q_\phi\left(\epsilon_t \mid z_{t-1}, U_t^{(s)}, U_t^{(p)}, s_t\right)\|p_\theta\left(\epsilon_t \mid z_{t-1}, s_{t-1}\right)\right]$$
$$- \text{KL}\left[q_\phi\left(z_t \mid z_{t-1}, U_t^{(s)}, U_t^{(p)}, s_{t-1}\right)\|p_\theta\left(z_t \mid z_{t-1}, s_{t-1}\right)\right] \tag{16}$$

The first loss term represents the reconstruction loss, while the latter two correspond to the KL divergence of the exogenous and endogenous variables from the approximate posterior distribution.

Since the acyclic nature of the causal graph, it is essential to incorporate acyclic constraints, which are defined as: $h(\tilde{\mathbf{A}}) = \text{tr}\left[(I + \tilde{\mathbf{A}} \circ \tilde{\mathbf{A}})^m\right] - m$. Furthermore, the negative loglikelihood loss is employed for the response loss, expressed as $\mathcal{L}_g = -\frac{1}{L}\sum_{l=1}^{L}\log(\rho(r_l|r_{<l}, \boldsymbol{x}))$. In summary, the total loss of our model is a summation of the four losses:

$$\mathcal{L} = -\mathcal{L}_{ELBO} + \mathcal{L}_g + \lambda h(\tilde{\mathbf{A}}) + \frac{c}{2}|h(\tilde{\mathbf{A}})|^2, \tag{17}$$

where $\lambda$ is the Lagrange multiplier and $c$ is the penalty parameter.

## 4 EXPERIMENTS

### 4.1 DATASETS

The emotional dialogue dataset, ESConv(Liu et al., 2021), is utilized to evaluate our proposed model. Each dialogue includes the seeker's situation and dialogue context, with each sentence from the

Table 1: The automatic evaluation result for both the baselines and our model on the ESConv dataset. †indicates the results are obtained from the paper (Bao et al., 2024), while other results are obtained from the original paper. - denotes there is no report in the work. ↑ represents the higher the value, the better the performance.

| Model | PPL↓ | B-1↑ | B-2↑ | B-3↑ | B-4↑ | D-1↑ | D-2↑ | R-L↑ |
|---|---|---|---|---|---|---|---|---|
| Transformer †(Vaswani et al., 2017) | 81.55 | 17.25 | 5.66 | 2.32 | 1.31 | 1.25 | 7.29 | 14.68 |
| MoEL †(Lin et al., 2019) | 62.93 | 16.02 | 5.02 | 1.90 | 1.14 | 2.71 | 14.92 | 14.21 |
| MIME †(Majumder et al., 2020) | 43.27 | 16.15 | 4.82 | 1.79 | 1.03 | 2.56 | 12.33 | 14.83 |
| BlenderBot-Joint(Liu et al., 2021) | 17.39 | 18.78 | 7.02 | 3.20 | 1.63 | 2.96 | 17.87 | 14.92 |
| MISC †(Tu et al., 2022) | 16.32 | 17.73 | 6.75 | 3.23 | 1.83 | 4.19 | 17.76 | 15.43 |
| PoKE(Xu et al., 2022) | 15.84 | 18.41 | 6.79 | 3.24 | 1.78 | 3.73 | 22.03 | 15.84 |
| GLHG(Peng et al., 2022) | **15.67** | 19.66 | 7.57 | 3.74 | 2.13 | 3.50 | 21.61 | 16.37 |
| KEMI(Deng et al., 2023) | 15.92 | - | 8.31 | - | 2.51 | - | - | 17.05 |
| TransESC(Zhao et al., 2023b) | 15.85 | 17.92 | 7.64 | 4.01 | 2.43 | 4.73 | 20.48 | 17.51 |
| FADO(Peng et al., 2023) | 15.72 | - | 8.0 | 4.0 | 2.32 | - | - | 17.53 |
| PAL †(Cheng et al., 2023) | 16.78 | 18.77 | 6.91 | 3.03 | 1.51 | 4.10 | 22.73 | 15.29 |
| MFF-ESC(Bao et al., 2024) | 16.43 | 20.64 | **8.87** | **4.81** | **2.98** | 5.34 | 22.18 | **18.83** |
| SCBG(Xu et al., 2024) | - | 12.74 | 5.51 | 2.87 | 1.66 | 5.05 | 24.48 | 14.67 |
| ChatGPT(1 shot) †(Zhao et al., 2023a) | - | 13.91 | 4.53 | 1.96 | 1.02 | **5.92** | **31.38** | 13.19 |
| LLaMA-7B(0 shot) †(Bao et al., 2024) | - | 0.99 | 0.52 | - | - | 4.79 | 2.00 | - |
| **CausalESC(ours)** | 16.33 | **20.72** | 8.58 | 4.27 | 2.38 | 3.33 | 16.14 | 17.39 |

supporter annotated with the corresponding support strategy. Following previous work(Tu et al., 2022), dialogues are truncated every 10 sentences to form dialogue samples, and the dataset is randomly divided into training, validation, and test sets in a ratio of 8:1:1. To provide conditional information, each seeker's sentence is annotated with 6 categories of emotion labels, in accordance with the methodology described in the paper(Zhao et al., 2023b).

## 4.2 EVALUATION METRICS

**Automatic Metrics.** For automatic evaluation, various metrics were employed to assess the text generated by the model. (1) Perplexity (**PPL**) was used to measure the overall quality of the generated responses. (2) BLEU-1 (**B-1**), BLEU-2 (**B-2**), BLEU-3 (**B-3**), BLEU-4 (**B-4**)(Papineni et al., 2002) and ROUGE-L (**R-L**)(Lin, 2004) metrics were utilized to evaluate the lexical and semantic aspects of the generated responses; (3) Distinct-1 (**D1**) and Distinct-2(**D2**)(Li et al., 2016) were applied to assess the diversity of the responses by measuring the proportion of unique n-grams in the generated responses.

**Human Evaluation.** Following previous work(Zhao et al., 2023b), three experts were recruited to interact with the model for manual evaluation. They were asked to rate the generated responses based on *Fluency*, *Identification*, *Empathy*, *Suggestion*, and *Overall* score. To ensure a fair comparison, the professional annotators were blinded to the source of the generated text.

## 4.3 BASELINES

We compare CausalESC with several state-of-the-art models: **Transformer**(Vaswani et al., 2017), **MT Transformer**(Rashkin et al., 2019), **MoEL**(Lin et al., 2019), **MIME**(Majumder et al., 2020), **Blenderbot-Joint** (Liu et al., 2021), **MISC**(Tu et al., 2022), **PoKE**(Xu et al., 2022), **GLHG** (Peng et al., 2022), **KEMI**(Deng et al., 2023), **TransESC**(Zhao et al., 2023b), **FADO**(Peng et al., 2023), PAL(Cheng et al., 2023), MFF-ESC(Bao et al., 2024) and SCBG(Xu et al., 2024). More details about these models are described in Appendix *Baselines*.

## 4.4 OVERALL RESULT

**Automatic Evaluation** The automatic results of our model are shown in Table 1. Compared with empathy response models (Transformer, MoEL, MIME), our model's performance is significantly improved. This improvement may be attributed to the fact that their training objectives are not

related to emotional support, making it difficult for them to handle challenging ESC tasks. Regarding BlenderBot-based models (MISC, PoKE, GLHG, KEMI, TransESC, FADO, PAL, MFF-ESC, SCBG), CausalESC does not use any external knowledge, yet its performance surpasses these external knowledge-enhanced methods. This demonstrates the model's ability to capture the dynamic evolution of the seeker's internal state under the supporter's strategy, improving the quality of generated responses. Additionally, compared to large models such as ChatGPT and LLaMA-7B, our model achieves promising results regarding B-n and R-L indicators. Although its performance is inferior to that of existing large models in terms of D-n, the higher diversity may lead to a larger deviation from the true responses.

**Human Evaluation** The results of the human evaluation, shown in the Tabel 2, indicate that CausalESC significantly outperforms BlenderBot-Joint and MISC. Compared to BlenderBot-Joint, our model excels in Fluency, Empathy, and Overall score, effectively demonstrating its ability to perceive the seeker's cognitive, emotional, and behavioral states and generate more empathetic responses. CausalESC still

Table 2: Human interaction evaluation results (%). Our model has a significant improvement with $p$-value $< 0.05$.

| CausalESC vs. | BlenderBot-Joint | | | MISC | | |
|---|---|---|---|---|---|---|
| | **Win** | **Lose** | **Tie** | **Win** | **Lose** | **Tie** |
| Fluency | **55.2** | 10.0 | 34.8 | **60.5** | 20.5 | 19.0 |
| Identification | **51.0** | 13.5 | 35.5 | **48.0** | 11.0 | 41.0 |
| Empathy | **53.0** | 7.0 | 40.0 | **55.3** | 15.5 | 29.2 |
| Suggestion | **49.7** | 12.3 | 38.0 | **47.0** | 20.0 | 33.0 |
| Overall | **59.0** | 10.2 | 30.8 | **57.0** | 16.0 | 27.0 |

achieves higher performance in all five aspects, even though MISC utilizes more external knowledge. This demonstrates that the temporal causal mechanism of the seeker's cognitive, emotional, and behavioral states is conducive to generating supportive responses.

## 4.5 Ablation Study

An ablation study, summarized in Table 3, illustrates the contribution of each component to the final result. First, removing the support intervention module lowered the automatic evaluation scores, underscoring the importance of the supporter's strategy in influencing the seeker's internal state. Second, deleting the CEB causal module led to a significant drop in performance,

Table 3: The evaluation results of ablation study on each module.

| Model | PPL↓ | B-1↑ | R-L↑ |
|---|---|---|---|
| CausalESC | **16.33** | **20.72** | 17.39 |
| w/o support intervention module | 17.27 | 18.94 | 16.41 |
| w/o CEB causal module | 17.85 | 19.35 | **17.54** |
| w/o damping module | 17.96 | 19.01 | 16.73 |
| w/ signal transfer | 17.17 | 20.51 | 17.31 |
| w hybird schema | 16.73 | 19.24 | 17.09 |

emphasizing the necessity of modeling the causal relationships among cognition, emotion, and behavior. Similarly, eliminating the damping module reduced scores. Then, we replaced the *multidimensional interaction transfer mechanism* with a *single transfer mechanism* resulting in a marked decline, demonstrating the former's effectiveness in capturing state interdependence. Finally, using the hybrid schema caused varying performance reductions for CausalESC.

## 4.6 In-depth Analysis

**Case Study** A case study is presented in Table 4, featuring responses generated by CausalESC and two state-of-the-art baselines. While the reaction from BlenderBot-Joint provides a suggestion, its hesitant tone undermines the guidance, resulting in a response that lacks enthusiasm (see blue phrase) and depth (see red phrase). In the case of MISC, the response offered clear affirmation; however, it was superficial and failed to explain the benefits of spending time with and caring for the dog (see red phrase). This lack of depth prevented the seeker from elaborating on the emotional or psychological advantages. In contrast, CausalESC's response affirmed the seeker's choice by highlighting Corgi's intelligence(see purple phrase). This demonstrates that our model can understand the seeker's genuine needs and provide effective responses, which stems from modeling the seeker's emotions, cognition, and behavioral transfer processes during each conversation round.

**Interpretability of Latent Variables** To evaluate the model's effectiveness in disentangling cognition, behavior, and emotion, we employed t-SNE(Van der Maaten & Hinton, 2008) to visualize the latent representation of the seeker's internal state at the first and last time step. The specific experi-

Table 4: Case study of the generated responses by CausalESC and other models. Due to limited space, we omitted some sentences.

| Dialogue Floor | |
|---|---|
| **Supporter** | Awl so sorry to hear that. There are times in all our lives that we feel alone. This feeling will change |
| **Seeker** | Thanks for the tip |
| **Seeker** | I was thinking about getting a dog. Is that a good idea? |
| | ...... |
| **Seeker** | Corgi, thoughts? |
| **Response** | |
| **BlenderBot-Joint** | i am not sure. i don't know how to go into more detail. but i think that you could find a really nice dog. (***Lacks the enthusiasm and depth***) |
| **MISC** | i think that would be a great choice. i think it would be great to have a dog that you can spend time with and care for. (***Superficial affirmation***) |
| **CausalESC** | i think that is a great choice.  i think you will be surprised at how intelligent a dog surprised at how intelligent a dog can be. (***Affirmative insight***) |
| **Ground truth** | Those use dogs are super smart as well. I think that's a great choice) |

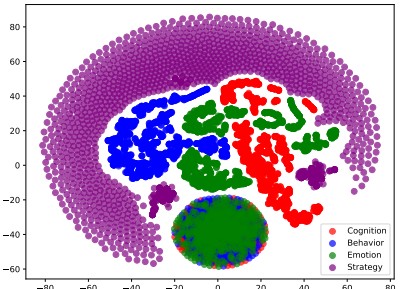 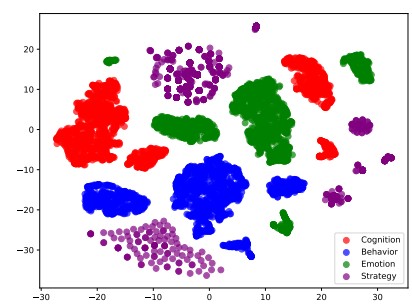

(a) Data distribution at the first time step  (b) Data distribution at the last time step

Figure 4: The t-SNE plot of cognition, emotion, behavior and strategy learned by ours.

ments are presented in the Figure 4. The result reveals that at the first moment, the clustering effect is poor, with low separation between the latent factors (emotion, cognition, behavior, and strategy). In contrast, at the last time step, the sepration between the factors is significantly improved, reflecting the model's dynamic learning capability during the dialogue process. This demonstrates that our model could decouple and dynamically represent complex emotional, cognitive, behavioral, and strategic factors.

## 5 CONCLUSION AND FUTURE WORK

This paper proposes a new model, CausalESC, for capturing the evolution mechanism of cognitive, emotional, and behavioral during the dialogue process. The key contribution lies in breaking the causal loop problem by assuming that the mutual influence of emotion, behavior, and cognition follows a first-order Markov property with *i.i.d.*. variables. Additionally, a support intervention module is proposed to consider the impact of the strategy on the seeker state, and a novel module is developed to capture the complex transfer process. Experimental results on both automatic and human evaluations show the superiority of our approach. In the future, we will explore more fine-grained states of the seeker, including physiological responses, and capture the evolution of these states.

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

## A    APPENDIX

Due to page limitations of the main body, the supplementary material includes theoretical proofs of acyclicity, additional implementation details and additional case studies that illustrate the following aspects:

## B    MORE METHOD DESCRIPTION

### B.1    ACYCLIC PROOF

To demonstrate that the interaction among emotion ($z_t^e$), cognition ($z_t^c$), and behavior ($z_t^b$) over time forms a DAG under the influence of independently and identically distributed (i.i.d.) random variables and an external perturbation ($\epsilon_t^{po}$).

**Markov Property Definition.** The Markov property states that the future state of a process depends only on its current state, not on the sequence of past states that preceded it:

$$P(X_{t+1}|X_t, X_{t-1}, \ldots, X_1) = P(X_{t+1}|X_t) \tag{18}$$

**Theorem.** Let us assume that the time series $B_t \in \mathbf{R}^m$ satisfies the following evolution equation:

$$B_{t+1} = f(B_t, \alpha_{t+1}) \tag{19}$$

where $\alpha_t \in \mathbf{R}^m$. Then, $B_t$ is a first-order Markov chain if and only if the sequence $\{\alpha_t\}$ is independently and identically distributed (i.i.d.)(see (Sun et al., 2015)).

**Proof.** Consider the latent variables $z_t = [z_t^e, z_t^c, z_t^b]$ is governed by:

$$\begin{aligned}
z_{t+1}^e &\sim f_e(z_t^e, z_t^c, z_t^b, \epsilon_t^{po}, \eta_{t+1}^e) \\
z_{t+1}^c &\sim f_c(z_t^e, z_t^c, z_t^b, \epsilon_t^{po}, \eta_{t+1}^c) \\
z_{t+1}^b &\sim f_b(z_t^e, z_t^c, z_t^b, \epsilon_t^{po}, \eta_{t+1}^b)
\end{aligned} \tag{20}$$

where $\eta_{t+1}^e$, $\eta_{t+1}^c$, and $\eta_{t+1}^b$ are i.i.d. random variables, and $\epsilon_t^{po}$ represents external perturbations. Since $z_{t+1}^e$, $z_{t+1}^c$, and $z_{t+1}^b$ evolve based on the past states and i.i.d. noise, the random sequences $\eta_{t+1}^e$, $\eta_{t+1}^c$, and $\eta_{t+1}^b$ are independently and identically distributed. According to equation 18 and equation 19, this implies that each $z_{t+1}^e$, $z_{t+1}^c$, and $z_{t+1}^b$ satisfies the first-order Markov property. Consequently, when considering the interaction across time steps, the variables $z_{t+1}^e$, $z_{t+1}^c$, and $z_{t+1}^b$ form a Directed Acyclic Graph (DAG) in the time dimension.

## C    MORE IMPLEMENTATION DETAILS

### C.1    EXPERIMENTAL SETUPS

For a fair comparison with previous work, we use the 90M BlenderBot(Roller et al., 2021) as the base model. AdamW (Loshchilov & Hutter, 2019) is employed as the optimizer, and training is conducted for 4 epochs with a batch size of 20. The learning rate starts at 2e-5 and incorporates a linear warmup over 120 steps. The latent variable has a dimension of 48, with the dimension of the cognitive, emotional, and behavioral factors each being 16. During the training process, we use the KL annealing method to mitigate the KL vanishing problem. Training and testing are performed on a single GeForce RTX 3090 GPU. For inference, the decoding algorithm utilized Top-p and Top-k sampling with parameters set to $p = 0.3$ and $k = 30$.

## C.2 BASELINES

We compare our proposed model with several state-ofthe-art models:

- **Transformer** Vaswani et al. (2017): This model is a Seq2Seq model based on the Transformer architecture.

- **MoEL** (Lin et al., 2019): A Transformer-based model that combines sentiment distribution representation from multiple decoders to enhance the empathy of generated responses.

- **MIME** (Majumder et al., 2020): Another transformer-based model employs emotion polarity and emotion mimicry to generate empathetic responses, while also introducing randomness into the emotion mixture to produce more diverse responses.

- **Blenderbot-Joint** (Liu et al., 2021): The Blenderbot model fine-tuned on the ESConv dataset, serving as a baseline model for this dataset. The generation strategy involves adding a special support strategy token at the beginning of the response to guide its generation.

- **MISC** (Tu et al., 2022): This model incorporates COMET to improve the understanding of the seeker's emotions and generate more supportive responses through mixed strategy representation.

- **PoKE**(Xu et al., 2022) Latent variables are utilized to represent the one-to-many relationship of support strategies.

- **GLHG** (Peng et al., 2022): A hierarchical graph neural network captures various information, including global reasons, local intentions, and conversation history, and builds hierarchical readability between them to generate emotional support response.

- **KEMI**(Deng et al., 2023) The model retrieves real-world case knowledge from a large-scale mental health knowledge graph to generate mixed-initiative responses.

- **TransESC** (Zhao et al., 2023b) The state transition graph network captures three types of turn-level transmission information: semantic transmission, strategy transmission, and sentiment transmission, facilitating effective dialogue generation smoothly and naturally.

- **FADO**(Peng et al., 2023) The model leverages turn-level and conversation-level feedback to penalize strategy and present context-to-strategy and strategy-to-context flow, generating responses.

- **PAL**(Cheng et al., 2023) The model employs personal information alongside a controllable strategy-based generation method to provide personalized emotional support.

- **MFF-ESC**(Bao et al., 2024) This paper propose a multi-stream information fusion framework that fully integrates the text semantic stream, sentiment intensity stream, and feedback stream to simulate the transformatioiin of sentiment intensity.

- **SCBG**(Xu et al., 2024) The model leverages semantic constraints during the generation process to produce supportive responses that are relevant to the user.

# D MORE EXPERIMENTAL RESULT

## D.1 CEB CAUSAL ANALYSIS

To explore the causal graph between the seeker's cognition, emotion, and behavior, experiments were conducted to analyze the learned causal graph, as shown in Fig.5. It was found that the causal graph of the seeker's internal state remains consistent across all samples and time steps, demonstrating a high degree of stability and consistency. Specifically, the cognitive state $c$ directly affects both the emotional state $e$ and the behavioral state $b$, while the behavioral state further influences the emotional state. This observation aligns with the conclusion on psychological mechanisms and response patterns, implying that CausalESC can learn psychological concept representations that conforms to true causal relationships.

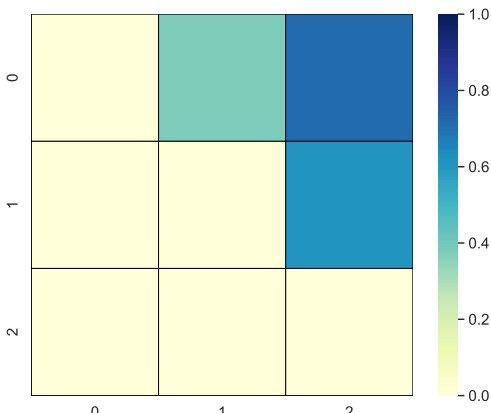

Figure 5: The learned causal graph $A$. The concepts include: 0 Cognition; 1 Behavior; 2 Emotion.

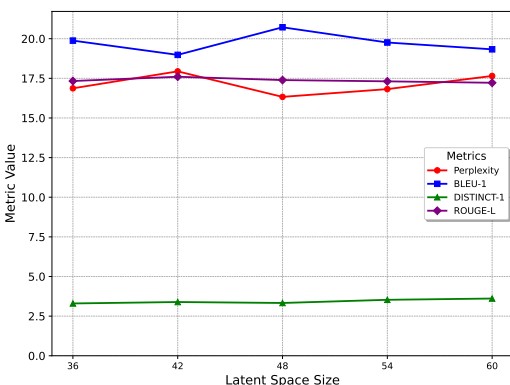

Figure 6: Effects of latent space dimensionality on the disentanglement of cognition, behavior, emotion and strategy.

## D.2 EFFECT OF HIDDEN SIZE

We evaluate the impact of varying latent space dimensions on the experimental results, as shown in Fig.6. Specifically, we test five configurations: 36, 42, 48, 54, and 60 dimensions. As the result indicates, the model performs best when the latent space is 48. We observe that larger latent spaces may lead to overfitting, while smaller spaces can restrict the model's capacity to learn. Therefore, careful selection of latent space dimensions is crucial for optimal performance. This further demonstrates the model's ability to capture the dynamic evolutions in the conversation.

