# OpenReview forum: "CausalESC: Breaking Causal Cycles for Emotional Support Conversations with Temporal Causal HMM"
_ICLR.cc/2025/Conference — ICLR 2025 Conference Withdrawn Submission_

### Official Review · Reviewer_qag5 · 2024-10-28

**Soundness:** 2
**Presentation:** 2
**Contribution:** 1
**Rating:** 3
**Confidence:** 4

**Summary:**

This paper builds on a current research program looking into "Emotional Support Conversation", using chatbots to provide psychological support. It constructs a model that uses several causally connected 'psychological' latent variables  that are presumed to relate to the seeker's generation of an utterance and used to generate the next appropriate utterance by the model. The authors also introduce several other knobs to their model, including for example a 'damping' module. The behavior of the model is compared to several existing models using auto-evaluation as well as human experts. In a head-to-head with two other models (BlenderBot-Joint and MISC), the current model is preferred by human experts. The authors also provide a specific test case.

**Strengths:**

The problem that the authors are overall attempting to address (increase in mental health issues combined with a limited support system) is timely and urgent.

The overall direction of examining latent 'psychological' variables is a welcome addition to the current landscape of support systems.

The work seems technically sound.

I appreciate the comparison to multiple other models, and the use of multiple evaluation methods.

**Weaknesses:**

This paper has several issues that prevent a recommendation for acceptance. Some of these are not uniquely the issue of this paper in itself, but more the eco-system within which the paper is written (ESC). I'll detail more below, but as one example, the opaque reliance on three 'human experts' to examine such tasks seems especially problematic, but is also used in other accepted papers that this paper relies on. Still, this seems like a negative practice such that precedence is not a good guide for it.

In more detail:

* While the paper makes general contact with psychology and cognitive science, it is superficial at best. Generally pointing at things like 'cognitive behavioral therapy' or 'emotional regulation' ignores most of the work that has been done to actually test, validate, and model these things. I'm exaggerating to make a point, but the situation is similar to trying to model the physics of a situation while ignoring what we know actually about physics, broadly citing a physics textbook, and then throwing an LLM at it that causally connects in a loop variables labeled things like 'friction', 'momentum', and 'heavy-ness' (some of those are indeed professional terms, some are not, and simply connecting them in a DAG doesn't do much).

* Looking at figure 2 for example, how exactly are things like 'emotion, cognition, behavior' actually meant to affect one another? There are huge literatures in computational cognitive science trying to connect these things that go into actual models of them, but here they're all just kind of in a soup together. You would think that 'behavior' would be something like an action variable that is determined by a person's cognitive and emotional state ("I am hungry, there is an apple over there, I will reach for the apple") but then the next state should be determined by the behavior as the actual causal variable, why would the state determine the behavior except through its affect on mental states, unless this is meant to be some sort of habit or instinct pathway?

* Continuing that line: The 'damping' module has a specific technical role, keeping the variables within some bound. That's presumably not meant to have a psychological reality, or if it does it is presumably something rather biological. Instead, the authors connect it to 'emotion regulation theory' which has a very specific, tested meaning within psychological and refers to specific strategies people use to either 1) reassess their current situation or 2) tamp down outward behavior expressing their emotion. Instead, the 'damping' is happening over cognition, emotion, behavior broadly without any attempt to model or validate the actual procedure of 'emotion regulation' in the well-accepted meaning of this term. I don't think the authors *should* have to validate that their damping module is in fact implementing something like a re-assessment strategy, there's no reason to expect that it is doing that, but then why call it that and connect it to something it isn't doing? I mention this point on its own but also as example of the overall weakness of the paper in connecting to cognitive/psychological work in general.

* The authors claim as one of the main contributions the unfolding of a causal loop into multi-step MDPs, but this is pretty standard in modeling how agents interact with the world using MDPs and latent mental variables (see e.g the work of Chris Baker and Josh Tenenbaum on inverse planning)

* I appreciate the comparison to multiple other models but it seems like the authors' model basically misses out on nearly all the comparisons. That is, for almost every metric there seems to be a better fit module, the authors mostly write around this rather than facing it head on.

* Probably the most problematic issue in the paper is the actual testing of whether this model does what it is purported to do. In this analysis the authors use 'three experts' but very little detail is provided about this evaluation. Who were these experts? In what way were they experts? What training did they receive and how did they do the evaluation? Were they students from the lab? Colleagues? Naive participants recruited online? Why was the comparison to two models that didn't do that well on the auto-eval instead of the models that did better than the authors model on the auto-eval (e.g. chatGPT)? Presumably if these systems are ever going to be released into the wild the relevant comparison would be to show the dialogs to a large and diverse sample of naive human participants and have them choose the preferred continuation, and not just between models but between models and a natural conversation. Otherwise, the models may do better than one another while existing in a completely closed-loop format that is light-years away from natural dialog.

* I appreciate the use of the examples and test cases but they only highlighted just how subjective the nature of 'expertise' judgement can be here. For example, I personally found the suggestion of the system in Figure 1 to be non-supportive and downright missing the pragmatics of the situation (if a friend told me they were worried about losing their job I might think they're trying to find comfort, and if I were to suggest to them 'hmmm, yeah, better update your resume' they might be taken aback, thinking that I thought they could in fact lose their job). As another example, I didn't find the "I think you will be surprised at how intelligent a dog can be" to be some kind of 'affirmative insight' that understands the seeker's 'genuine needs',  nor do I find the previous example to be 'superficial affirmation'; this all feels like reading tea-leaves.

**Questions:**

* Can the authors provide significantly more detail about how the human-based evaluation worked?

* What would happen to the authors' results and conclusions if instead of '3 experts' the authors used a diverse and large group of naive participants tasked with assessing which continuation they preferred?

* What would happen to the authors' results and conclusions if they compared their model using human experts beyond BlenderBot-Joint and MISC, to other models including the ones that did better on the auto-eval, and to human-generated response?

---

### Official Review · Reviewer_hAFe · 2024-11-03

**Soundness:** 3
**Presentation:** 2
**Contribution:** 3
**Rating:** 6
**Confidence:** 2

**Summary:**

This paper introduces CausalESC, a model designed to improve Emotional Support Conversations (ESC) by dynamically representing a support seeker’s evolving internal states (cognition, emotion, behavior) and adapting responses accordingly. The model employs a Markov process to handle the causal dependencies among emotions, cognition, and behavior over time. It also comprises a prior and posterior network to disentangle seeker’s internal states with the support strategy factors. An additional damping transfer mechanism stables the interactions between internal states and strategy. Experimental results show that CausalESC demonstrated improved empathy, fluency, and response quality over baseline models on the ESConv dataset in both automatic and human evaluations.

**Strengths:**

1. CausalESC models the evolution of a seeker’s internal states (emotion, cognition, and behavior) over time, which is a significant improvement over previous methods that often treat these states as static snapshots.
2. Unlike some models that use external knowledge bases for empathy and support strategies, CausalESC achieves high performance with its internal mechanism alone. This independence reduces dependencies on external resources and ensures a more generalized approach.

**Weaknesses:**

Some notations are confusing:
- The upper subscription $s$: some are used for seekers and others are used for strategy.
- In Equation 2, the notation $\le T$ suggests cumulative variables up to time $T$. However, in other places, such as equation3, variables are simply indexed by $t$.
- $z_{po}$ and $z_{pr}$ are heavily used without a clear definition.

The comparison with the BlenderBot-based models in Table 1 is not significant. I think the proposed method appears to be compatible, rather than exclusive, with the BlenderBot-based models. It can still incorporate external knowledge into the decoder and achieve better results.

**Questions:**

What is the hybrid schema in Table3?

---

### Official Review · Reviewer_dkMZ · 2024-11-05

**Soundness:** 3
**Presentation:** 2
**Contribution:** 2
**Rating:** 3
**Confidence:** 4

**Summary:**

The paper presents "CausalESC," a novel model for Emotional Support Conversation (ESC) tasks, designed to alleviate emotional distress by capturing the evolving internal state of a seeker in conversation. The model proposes a Temporal Causal Hidden Markov Model (HMM) to represent the dynamic interplay of emotions, cognition, and behavior over time, grounded on a first-order Markov assumption. It includes a prior and posterior network that disentangle the seeker’s internal states and support strategies, alongside a damping transfer mechanism to regulate interactions among these components. Extensive experiments on the ESConv dataset show that CausalESC outperforms state-of-the-art ESC models, demonstrating its ability to provide responsive and supportive dialogue.

**Strengths:**

1. Dynamic Representation: By leveraging a Markov property, the model effectively captures the evolution of internal states across conversation rounds, addressing limitations of static models in ESC.
2. Interpretability: The model’s architecture, with distinct modules for cognition, emotion, behavior, and strategy, allows for a clear understanding of how the model interacts with different aspects of the seeker's internal state.

**Weaknesses:**

1. Complexity and Computational Overhead: The model introduces significant complexity, including multiple components like prior and posterior networks, a support intervention module, and a damping transfer mechanism. This could lead to substantial computational costs and may not be feasible in real-time applications.
2. Limited Benchmark Comparison: While the paper compares the model against several ESC baselines, it lacks a more comprehensive comparison with larger models (e.g., fine-tuned ChatGPT or LLaMA) on multiple ESC datasets, which could better contextualize its performance.
3. Grounding problem: How do you ensure that the hidden state corresponds to cognitive state, affective state, behavioral state ....?
4. Unclear writing:
* In line 209, what is the issue of circularity, could you offer some examples?
* The difference between $z$ and $\epsilon$ is not clearly stated in sec3.3
* In line 238, the statements 'the second term' , and 'the third term' should be written more clearly.
* In sec 3.3.1, the authors should write more on the holistic stuff, such as how the posterior, the prior ... work together to gain the next generated utterance
* In line 270, 'According to .. mechanism', the authors should offer references regarding this statement.
* The authors should provide more explanation on the designation of Eq7
* Is the psychocausal memory decoder a transformer decoder?
* In sec 4, the authors should talk about the training data, as this part is missing through out the paper.
* In table 1, the authors should add a row to demonstrate the gap between their work and the SOTA or the second-best model.
* In figure 4, what is the strategy factor?

5. Unsatisfying Performance: Table 1 shows that the performance is unsatisfying. Table 3 shows that what line 465-line 468 states is inaccurate, as R-L rises after deleting the CEB causal module.

**Questions:**

As seen above

**Details Of Ethics Concerns:**

The authors focus on psychological problems, and in sec4.4, human evaluation is involved. The authors should provide whether they have the permission to perform these human evaluation experiments.

---

### Official Review · Reviewer_BEMh · 2024-11-05

**Soundness:** 2
**Presentation:** 3
**Contribution:** 2
**Rating:** 5
**Confidence:** 3

**Summary:**

This paper proposes CausalESC, a temporal causal hidden Markov model for emotional support conversations that aims to capture the dynamic evolution of seekers' internal states. The key innovation is modeling the mutual influence between cognition, emotion, and behavior as a first-order Markov process with i.i.d. variables, which breaks potential causal cycles into a directed acyclic graph (DAG). The model consists of three main components: a dialogue floor encoder, a temporal causal hidden Markov module, and a psychocausal hybrid decoder. The authors claim their approach is the first to learn causal representations within causal loops in emotional support conversations.

**Strengths:**

1. The paper addresses an important challenge in emotional support conversations by modeling the dynamic nature of psychological states rather than treating them as static snapshots.
2. The theoretical foundation drawing from Cognitive Behavioral Therapy (CBT) provides grounding for the model architecture.
3. The proposed solution to break causal cycles using temporal unfolding and Markov assumptions is innovative and mathematically sound.
4. The model architecture is comprehensive, incorporating multiple relevant components like strategy intervention and holistic damping transfer mechanisms.

**Weaknesses:**

1. The evaluation section is missing from the provided content, making it impossible to assess the empirical validity of the claims.
2. The mathematical formulation lacks sufficient detail about how the holistic damping transfer mechanism works and how it ensures changes remain within reasonable ranges.
3. While the paper claims to be the first to learn causal representations within causal loops, it does not thoroughly discuss or compare with other potential approaches to handling circular causality.
4. The paper does not adequately discuss the limitations of the first-order Markov assumption, which may be an oversimplification for complex psychological processes.
5. The implementation details of the support intervention module using attention mechanisms need more elaboration on why this particular approach was chosen.

**Questions:**

1. How do you justify the i.i.d. assumption for psychological variables that are likely to be highly correlated across time steps?
2. What metrics were used to evaluate the "reasonable range" of changes in the holistic damping transfer mechanism? How were these thresholds determined?
3. How does the model perform when dealing with long-term dependencies that might violate the first-order Markov assumption?
4. Can you provide empirical evidence that the temporal unfolding of causal cycles actually captures the true psychological dynamics better than alternative approaches?
5. How computationally intensive is this model compared to existing approaches, given the additional complexity of temporal causal modeling?

---

### Note · Authors · 2024-11-25

I have read and agree with the venue's withdrawal policy on behalf of myself and my co-authors.